# Learning from the Past: Pandemics and the Governance Treadmill

D. G. Webster [1,*], Semra A. Aytur [2], Mark Axelrod [3], Robyn S. Wilson [4], Joseph A. Hamm [5], Linda Sayed [6], Amber L. Pearson [7], Pedro Henrique C. Torres [8], Alero Akporiaye [9] and Oran Young [10]

1 Environmental Studies, Dartmouth College, Hanover, NH 03755, USA
2 Department of Health Management and Policy, University of New Hampshire, Durham, NH 03824, USA; semra.aytur@unh.edu
3 Department of Fisheries & Wildlife, James Madison College, Michigan State University, East Lansing, MI 48825, USA; axelrod3@msu.edu
4 School of Environment and Natural Resources, The Ohio State University, Columbus, OH 43210, USA; wilson.1376@osu.edu
5 Environmental Science & Policy Program, Department of Political Science, School of Criminal Justice, Michigan State University, East Lansing, MI 48824, USA; jhamm@msu.edu
6 James Madison College and College of Human Medicine, Michigan State University, East Lansing, MI 48825, USA; sayedlin@msu.edu
7 Department of Geography, Environment, and Spatial Sciences, Michigan State University, East Lansing, MI 48825, USA; apearson@msu.edu
8 Divisão Científica de Gestão, Ciência e Tecnologia Ambiental, Instituto de Energia e Ambiente (IEE), Universidade de São Paulo, São Paulo 05508-010, SP, Brazil; phcampellotorres@gmail.com
9 Department of History, Philosophy, and the Social Sciences, Rhode Island School of Design, Providence, RI 02903, USA; aakporia@risd.edu
10 Bren School of Environmental Science & Management, University of California, Santa Barbara, CA 93117, USA; young@bren.ucsb.edu
* Correspondence: d.g.webster@dartmouth.edu

**Abstract:** Global human health threats, such as the ongoing COVID-19 pandemic, necessitate coordinated responses at multiple levels. Public health professionals and other experts broadly agree about actions needed to address such threats, but implementation of this advice is stymied by systemic factors such as prejudice, resource deficits, and high inequality. In these cases, crises like epidemics may be viewed as opportunities to spark structural changes that will improve future prevention efforts. However, crises can also weaken governance and reinforce systemic failures. In this paper, we use the concept of the governance treadmill to demonstrate cross-level dynamics that help or hinder the alignment of capacities toward prevention during public health crises. We find that variation in capacities and responses across local, national, and international levels contributes to the complex evolution of global and local health governance. Where capacities are misaligned, effective local prevention of global pandemic impacts tends to be elusive in the short term, and multiple cycles of crisis and response may be required before capacities align toward healthy governance. We demonstrate that this transition requires broader societal adaptation, particularly towards social justice and participatory democracy.

**Keywords:** pandemics; governance treadmill; crisis rebound effect

## 1. Introduction

During the COVID-19 pandemic, many are asking how we got here, how we might have prevented or at least mitigated this global disaster, and how we can better prepare for the next pandemic. A number of authors have already provided in-depth analyses of the response to COVID-19 and compared this crisis to pandemics of the past [1–7]. It is too early to say what long-term effects COVID-19 will have on public health governance—or broader political economic systems—but these impacts will certainly vary widely around

the world. The World Health Organization (WHO) recognizes that political and social legitimacy are critical to its successful implementation of the Sustainable Development Goals, and "such legitimacy usually requires public acceptance of the importance of the regulatory framework for economic and social development and a feeling of trust that regulation, implementation and enforcement will be conducted equitably, fairly, transparently and in the best interests of the public" [8]. Governance is explicitly recognized in two Sustainable Development Goals, which are: (i) SDG 16-Promote peaceful and inclusive societies for sustainable development, provide access to justice for all and build effective, accountable and inclusive institutions at all levels and (ii) SDG 17-Strengthen the means of implementation and revitalize the global partnership for sustainable development [9]. This raises interesting questions about the role of crises in the evolution of governance across scales and levels of analysis. In particular, how does experience with global crises like pandemics help or hinder the development of institutions (or formal and informal rules and norms [10]) at multiple levels that account for the complexity of systemic risks such as global public health threats [11]?

In this paper we start to answer that question by using the concept of the governance treadmill [12] to unpack the temporal dynamics of crisis response, particularly whether responses to a specific crisis will generate temporary changes in governance that quickly erode once the crisis has passed or will lead to healthy governance through lasting changes that prevent or mitigate future crises. Existing scholarship considers how local conditions interact with the impacts of phenomena such as globalization and international law. However, that literature tends to focus primarily on local economic structures [13] or government types [14,15]. We argue that learning from crises is constrained by structural inequities and societal responses that reinforce counterproductive problem narratives. We demonstrate how resources, understanding, and incentives can be successfully aligned towards threat prevention when these power disconnects and problematic narratives are addressed.

The approach is based in interdisciplinary synthesis that draws on diverse literatures including epidemiology, political economy, international relations, environmental justice, and systems analysis. From the public health literature, we already have an understanding of the capacities required for effective prevention or mitigation of public health threats like COVID-19 [4]. Furthermore, the World Health Organization [8] recognizes that the public health system may provide a unifying platform through which to facilitate policy coherence and support United Nations Member States in the implementation of the Sustainable Development Goals and the 2030 Agenda for Sustainable Development [9,16] to improve threat prevention. We will review these lessons, demonstrating that among the three broad categories of capacities (resources, understanding, and incentives), each is necessary, but none is sufficient on its own for fostering effective governance of global human security threats like pandemics. Our primary focus is on the factors, or precursors to governance, that either foster or hinder alignment among those capacities toward prevention.

Throughout, we draw examples from the history of cholera (*Cholera morbus*). Only two strains of cholera have potential for epidemic spread. These have caused seven pandemics to date, as shown in Figure 1 [17,18]. Given space constraints, we mainly focus on comparison between governance of cholera in New York and London, and then link these local-level responses to global public health governance. London is an obvious choice for illustration because it is well-known in public health as the place where John Snow first applied epidemiologic methods in 1848 and again in 1854 to characterize cholera cases by person, place, and time. This enabled him to test the hypothesis that water could serve as a vehicle for transmission. Although his research was not widely accepted until much later, his work identified cholera as a water-borne disease [19,20]. New York was then selected as a point of contrast. Although it followed a fundamentally different approach to managing the pandemic, it ultimately overcame the disease at about the same time as London.

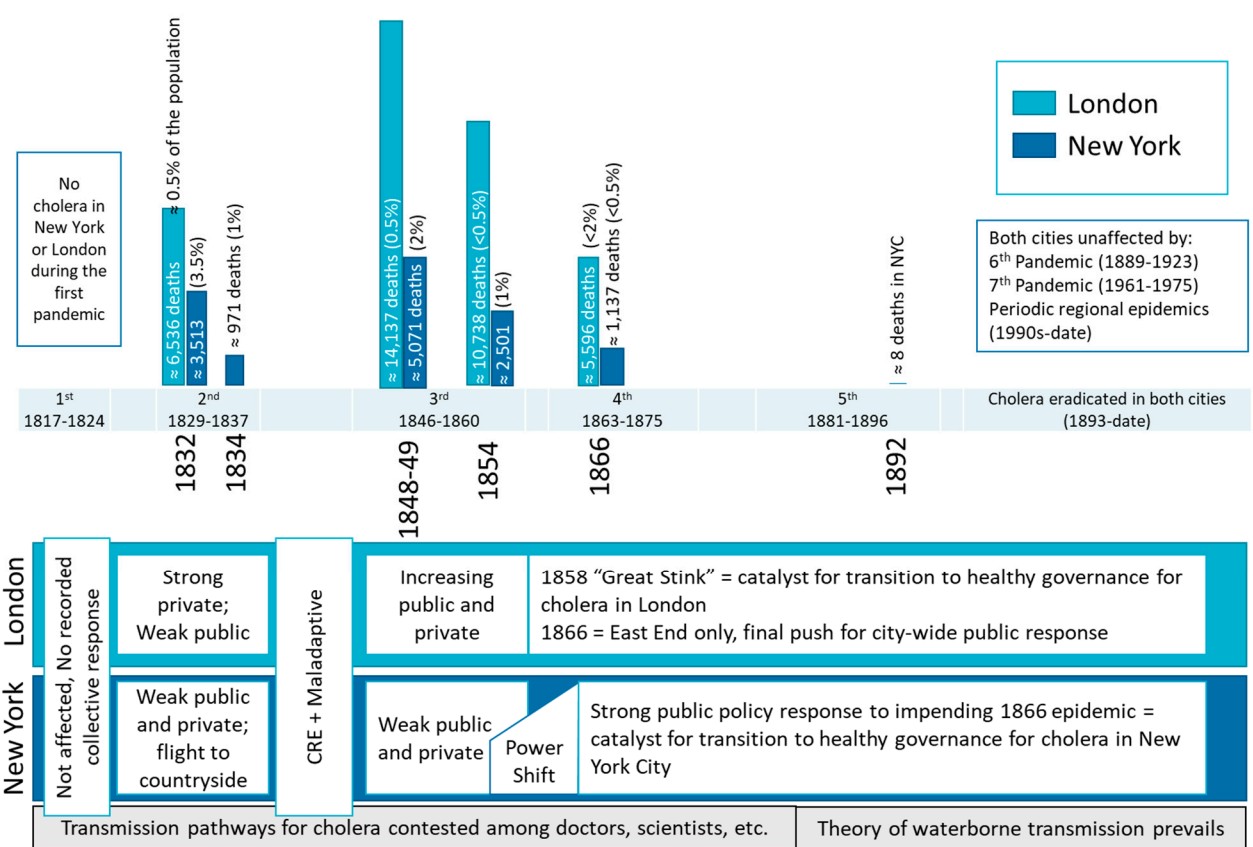

**Figure 1.** Cholera in London and New York. Health threat is represented by Total and (Per Capita) Mortality from Cholera. Sources: Mortality: London [21]; New York [22]. Population: [23,24]. Response is summarized from Sections 2 and 3.

As shown in Figure 1, both cities faced repeated epidemics as waves of cholera swept the world during the 1800s. Initial responses were insufficient in both cities, though a much smaller percentage of the population died in London during the 1832 epidemic. Total mortality was even higher during the epidemics of the 1840s, though population growth meant that mortality rates declined. Starting in the 1850s we see decreases in mortality and both cities finally conquered cholera in different ways by the end of the 1860s. Interestingly, these cities learned to cope with cholera through repeated experience well before germ theory was recognized, and indeed, before the theory of water-borne transmission of cholera was widely accepted in the 1870s [18]. However, if we take a wider view, while healthy governance to prevent cholera in these cities spread throughout the developed world, this in itself reduced incentives to fight the disease globally, and many still suffer from cholera today, whether at endemic or epidemic levels.

These patterns in the spatial and temporal dynamics of social change associated with epidemics can be observed in other modern and historical cases of response to epidemics [7,21]. Indeed, we would hypothesize that our framework applies to syndemic threats [22–24], in which infectious disease epidemics are synergistically overlaid on top of ongoing chronic disease epidemics, along with 'force multipliers' such as climate change and systemic racism [25].

The main difference is that infectious disease epidemics—or potential epidemics—are often acute, visible, and can affect large portions of the population, thus sending strong signals which may generate greater and quicker governance changes than other issues (at least in the short term). Detangling the complex, multi-level processes of healthy governance is difficult, but it yields understanding of its critical precursors [26]; not just what we need to change but what people can change under a given set of conditions.

While public health professionals are keenly aware of these limitations and have devised multiple methods to address them, effective prevention of hazards requires much broader change, including public acceptance of new norms regarding health care governance and increased willingness to hold ourselves and our representatives accountable for their decisions regarding public health, not just in the midst of the pandemic but also in its aftermath and beyond.

The paper is organized as follows. Section 2 starts by explaining the governance treadmill, a model for understanding feedback between threat perception, public concern, and public policy. Section 3 then describes how the three capacities of resources, understanding, and incentives need to align for prevention or mitigation of public health threats. Each of the three capacities is necessary, but insufficient on its own, to prevent the worst local impacts of global health threats. The section further shows how different types of problem narratives and power disconnects can either facilitate or prevent alignment of all three capacities as a temporary response to impending or ongoing crisis. Section 4 then examines how these three capacities change through the operation of the governance treadmill. This section starts with lessons about the crisis rebound effect and changes in the precursors for governance at the local level but then expands to discuss cross-level and cross-scale implications. As we conclude in Section 5, the pressures created by pandemics, combined with the many other underlying crises ranging from climate change to political upheaval, may foster new problem narratives that target systemic causes such as racism and global inequality. Indeed, these crises will be transformed into opportunities if they can trigger efforts to narrow power disconnects, ensuring that the people who have the greatest incentives to prevent such crises also have the resources and understanding to do so.

## 2. The Governance Treadmill

The *governance treadmill* is a conceptual approach designed to apply to a wide range of policy "problems", including public health threats like pandemics but also extending to environmental issues and other threats to human security. Here, "governance" covers the nexus of government, economy, and society, so it includes economic entrepreneurship as well as social institutions and public policy [27]. As will be seen throughout the text, all three of these pillars of governance must be considered to fully understand the temporal dynamics of pandemic response. Through this lens, we trace the effects of pandemics through multiple levels of analysis, from individual decision making through group behaviors in society and markets, to political activities, public policies, and, ultimately, the resulting effects of these collective responses on the initial problem (pandemic). This is why we use the generic term "people" except when individuals are acting in their specific roles as consumers, producers, policy makers, and so on. We consider that all forms of governance are "responsive" insofar as people are responding to signals about the nature of the problem, the behaviors of others, constraints on their own behavior, available solutions, and other factors in a heuristically rational way.

Figure 2 describes a generic governance treadmill for public health threats like pandemics. It is a starting point for discussions of the range of potential treadmill dynamics in subsequent sections. As the threat increases (e.g., disease spreads; top left box in Figure 2), its effects are felt more widely by more people (e.g., more infections; mortality and morbidity increase; more gossip and news stories about infections and deaths), which leads more individuals (e.g., consumers, producers, decision makers, etc.) to take actions to protect themselves and those they care about (e.g., improving sanitation at home and through charitable contributions to others, quarantining, demanding government assistance). These individual actions then culminate in collective response through social, economic, and political mechanisms such as sharing information about disease prevention, creating new informal social norms to avoid spreading the disease, aggregate changes in sanitation facilities due to individual charitable contributions, investing in efforts to find a vaccine, and establishing or implementing public policies to foster all of these individual-level changes.

Individual responses can be "productive" if they increase the effectiveness of collective response, but they can also be "counterproductive" if they reduce the effectiveness of collective response (e.g., spreading misinformation, refusing to wear masks, blocking effective policies). Depending on the composition of individual responses, collective response may be more or less effective in reducing the health threat.

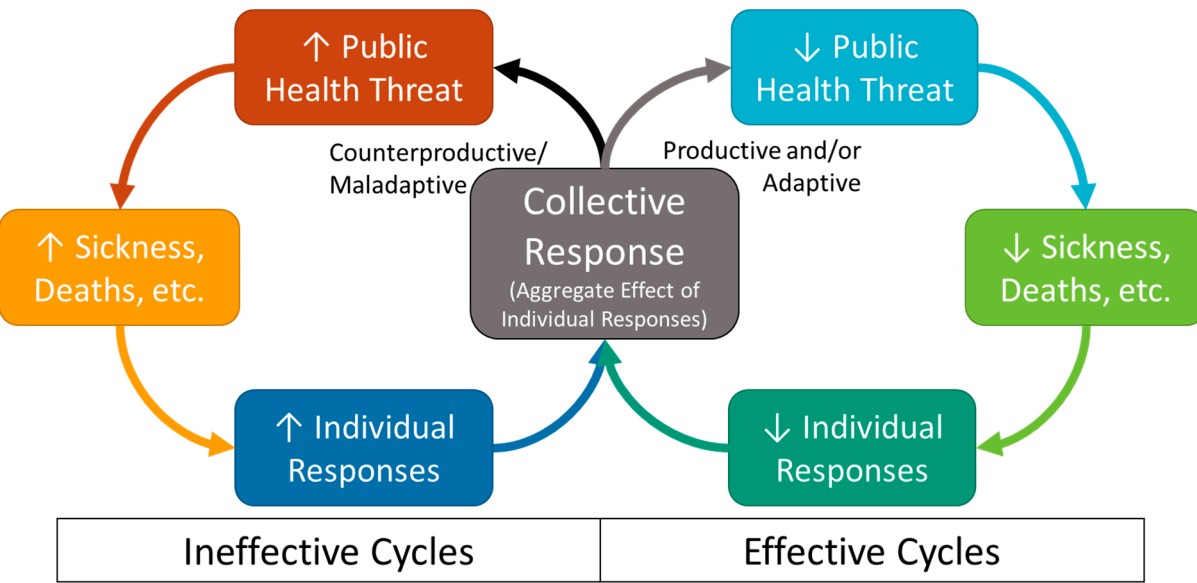

**Figure 2.** Governance Treadmill for Public Health Crises.

Collective response is "effective" when it is sufficient to reduce the public health threat (as measured by decreased mortality, morbidity and spread of disease), thereby switching the system to the right-hand side of the treadmill. It is important to keep in mind that effective collective responses may only be temporary. Indeed, as Figure 2 shows, when the treadmill is in effective cycles, the health threat declines, threat signals decrease, and therefore productive individual level responses decrease while the potential for counterproductive responses increases. This is known as the *crisis rebound effect*. If unchecked, the crisis rebound effect will lead to a return to ineffective cycles once the immediate crisis is over. Collective responses are "adaptive" if they include the creation of institutions that counteract the crisis rebound effect, keeping the treadmill on the effective side, and thereby preventing or mitigating future crises. Collective responses can also be "maladaptive" if they create institutions that keep the treadmill stuck in ineffective cycles. Importantly, collective responses that are "effective" in a given crisis may be "maladaptive" in the long term. Although our definitions of "effectiveness" and "adaptiveness" do not consider fairness and equity per se, research and our own analysis consistently show mitigation of public health threats is not possible without addressing the impacts on the most marginalized groups within society [28,29].

The governance treadmill produces different patterns for different problems in different contexts, as complex dynamics play out [30]. These patterns vary for many reasons, but it is useful to examine some stylized patterns that can occur as the treadmill cycles back and forth between ineffective and effective sub-cycles. At one extreme, if people collectively fail to learn from experience and consistently implement maladaptive responses, the resulting pattern of escalating crisis followed by stagnation or collapse is called "destructive governance" (Figure 3a). At the other extreme, adaptive collective response leads to a pattern of "healthy governance," which ensures that solutions are maintained over the long run (Figure 3c). Many different patterns can exist between these two extremes, but responsive governance usually leads to some variation on Figure 3b, where the treadmill switches back and forth between the ineffective and effective cycles in response to short-term signals. As a problem increases, people may try easy options first, or they may need time to learn about

the effectiveness and/or feasibility of their options. Either way, inaction or less effective responses tend to occur early on. Thus, the system may go through several ineffective cycles before switching into a period of more effective governance. Because response in this example is "effective" but not "adaptive", collective response dissipates as the problem recedes and the system will switch back to ineffective cycles fairly quickly.

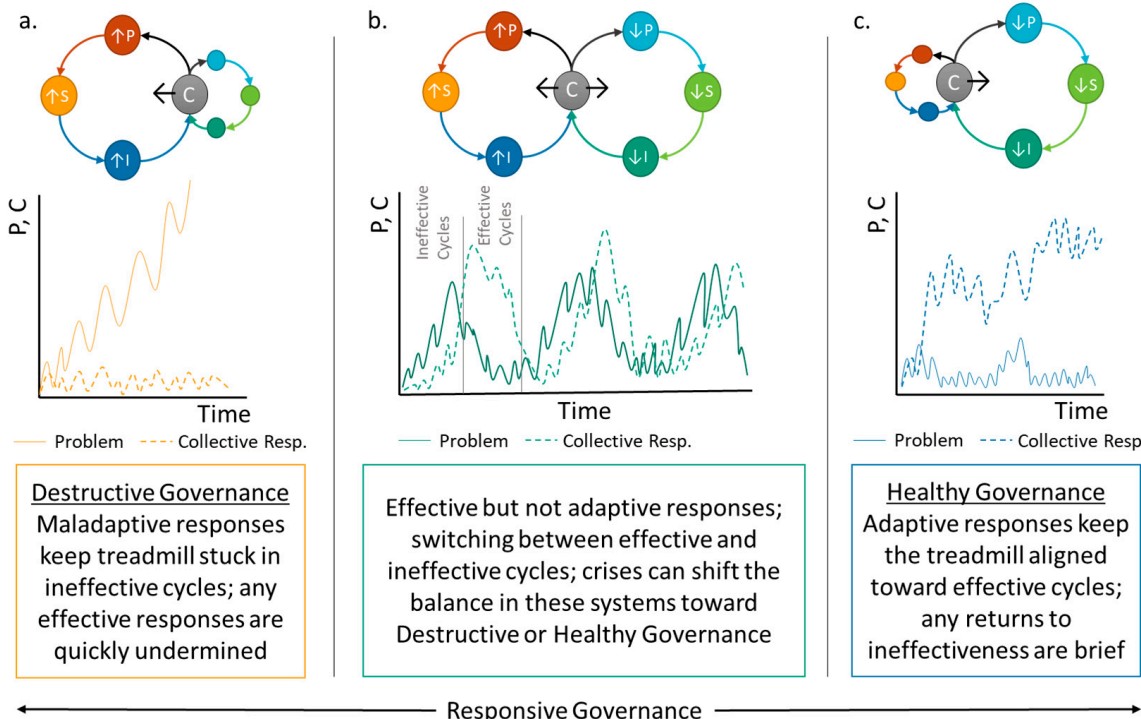

**Figure 3.** Stylized Patterns of Governance created through Treadmill Dynamics: (**a**) Destructive Governance Pattern, (**b**) Typical Governance Pattern, and (**c**) Healthy Governance Pattern.

From this framework, we can see that the term "treadmill" has multiple implications. Metaphorically, it reflects the apparent stagnation that is often observed as governance systems at multiple levels get mired in ineffective governance or when they cycle back and forth between more or less effective governance without actually "solving" governance problems (Figure 3a,b). On the other hand, while a person who is using a (physical) treadmill may be staying in the same place, their bodies are adapting because of the exercise. Just as consistent use of a real treadmill can help keep an individual healthy, consistent, adaptive use of the governance treadmill leads to healthy governance (Figure 3c). In the next section, we look at how the alignment of three major capacities facilitates effective collective response, causing a system to switch from the left to the right side of the treadmill during a specific crisis event. We will then examine how experience with multiple crises may move the system beyond effective response to adaptive response, initiating a transition to healthy governance in Section 4.

## 3. Responding to a Specific Crisis

In this section, we synthesize multiple disciplinary perspectives to understand how changing capacities can align a system toward more effective collective response during a specific human security crisis—causing a temporary switch from ineffective cycles to effective cycles in the treadmill. This analysis includes insights from epidemiology and public health that recognize that effectively preventing or mitigating a public health crises like the COVID-19 pandemic in 2019–2022—with anticipated lingering effects beyond—requires addressing a wide range of factors to limit the incidence of disease in a population, minimize the progression and/or transmission of disease, and reduce the negative conse-

quences of disease [31]. For example, important elements of different societies' prevention of epidemics/pandemics include: developing vaccines and altering conditions that affect the emergence of disease (primary prevention); changing behaviors and community conditions that limit transmission (secondary prevention/mitigation), and enabling critical health system capacity (e.g., adequate ICU beds, staffing, therapeutics; for tertiary prevention) [32,33].

Despite this wealth of knowledge about disease prevention, effective collective response tends to be elusive, as we have seen in the slow and sometimes counterproductive responses to COVID-19 around the world [2]. Drawing on the wider governance literature can elucidate how treadmill dynamics during a crisis alter the three key capacities for effective governance as described in Section 3.1. After explaining these capacities, we introduce the linked concepts of problem narratives and power disconnects, which together direct the alignment of these capacities to either keep the system stuck in ineffective cycles for another round or switch the system to more effective cycles. Section 3.2 provides examples of these dynamic local responses to global human security crisis, drawing on evidence from the cholera epidemics in New York and London in 1832.

### 3.1. Aligning Capacities for Effective Governance

From the scholarly literature, we identify three interdependent capacities that must align for a temporary transition from ineffective cycles to effective cycles of collective response to a given crisis: resources, understanding, and incentives (Figure 4). By "alignment", we mean that all three are necessary; if one is set in a counterproductive direction, then response will be ineffective, and the system will stay on the left-hand side of the governance treadmill for another cycle. First, everyone must have access to sufficient *resources* for prevention or, at least, mitigation. These resources include physical capital (e.g., hospitals, manufacturing plants), financial capital, social and political influence/autonomy (e.g., trust, social networks, etc.), and psychological capacity [34,35]. Second, people need a sufficient *understanding* of the problem and potential solutions to guide their effective use of those resources. Most importantly, they need to recognize their personal threat from the disease and sufficiently understand its causes to accurately identify effective and feasible solutions [36,37]. Understanding depends heavily on past experience [38], the way information is processed [39], social networks and norms [40], media representations [41–43] and many other factors described in greater detail below. Third, people must have *incentives* to actually use their resources effectively. Avoiding harm from the disease is an important incentive, but this is balanced against the perceived costs and benefits of various solutions, some of which allow individual avoidance without benefit to others [44]. This calculation is often undertaken intuitively and is influenced by emotional factors such as loss of personal identity and social factors such as the perceived fairness of outcomes, as well as economic factors such as gains or loss in utility [39,45–52].

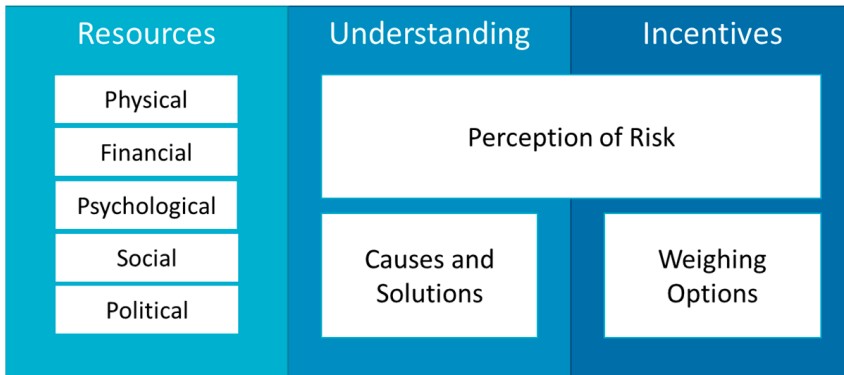

**Figure 4.** Capacities Required for Crisis Prevention or Mitigation.

These capacities can be influenced by many cross-level interactions; here we focus on problem narratives and power disconnects. When considering complex problems like public health crises, people rely on *problem narratives* to explain the issue as it relates to their own capacities, the shared capacities to which they have access through society or government, and the state of the world more broadly [53–55]. These narratives evolve as people receive and process new information about the problem and potential solutions (understanding), their private and collective response options (resources), and their own preferences, social norms, or government requirements (incentives). However, most problem narratives are embedded in broader conceptual narratives and ideological beliefs about the state of the world and the appropriateness of behaviors within it (e.g., ethics, values, political identity, and religion). These broader narratives may limit the types of information that people will accept or the types of responses considered feasible, thereby limiting people's understanding of a public health threat. However, people also use problem narratives to justify how resources are used and the ordering of conflicting incentives, so narratives may be shaped by existing capacities through various heuristics, biases, and motivated reasoning as described in greater detail in Section 4.2 [39,56].

Narratives do not have to be especially accurate or detailed to encourage effective collective response. Even under conditions of great uncertainty or lack of detailed knowledge of a disease, people may be able to construct effective problem narratives if they are able to learn from observation and experience [57–59]. However, narratives that under- or over-represent risk, or which prescribe maladaptive responses, can be considered counterproductive because they undermine effective collective response [60–62]. For instance, some problem narratives target symptoms or correlates rather than causes of a disease, keeping the treadmill stuck in ineffective cycles. Narratives that discount scientific evidence or willfully misinterpret evidence for personal gain can be the most pernicious [63–67]. Although it may not always be possible to know which problem narratives are effective until after the fact, effective narratives are often promoted through the acceptance of scientific evidence and the effective use of observations that underly traditional knowledge collected by indigenous peoples [68–70].

Because people have different capacities and believe different problem narratives, collective response to an impending crisis will depend heavily on *power disconnects*, which occur when people who have access to resources lack the understanding or incentives to use them effectively (i.e., they believe in maladaptive problem narratives) while those who have understanding and incentives to prevent disease (i.e., they believe in adaptive problem narratives) lack resources to do so [12,71,72]. Power disconnects are most often associated with the inequitable distribution of resources, but even in a highly unequal society the equitable distribution of perceived risk can provide incentives for powerful people to take preventative action, keeping power disconnects narrow [12,73–75]. Nonetheless, when powerful people can use their resources to insulate themselves from negative effects of a public health threat, they can mitigate their own risk while failing to protect others. Such signal-reducing responses are often rationalized through modifications of problem narratives and are one of the main reasons why a social system may remain on the ineffective side of the governance treadmill in spite of repeated experience with the same type of public health crisis. Indeed, power disconnects and problem narratives tend to be co-constructed as people try to make sense of the health threat within broad, pre-existing social narratives about medicine, religion, race, class, gender, and the role of government, to name a few [76]. We will return to these cross-level interactions in Section 4.

### 3.2. Illustrating Alignment Successes and Failures

Here we illustrate the points above by comparing the responses of London and New York during the 1832 cholera epidemics. Information in this section was drawn from a set of historical references about experiences with cholera in London [77–81] and New York [82–85]. Prior to the initial outbreaks of cholera in 1832, both cities had sufficient physical, financial, and technological resources to prevent the disease from reaching epi-

demic levels. Both cities had highly regarded private hospital systems and over-worked, under-resourced public hospital systems. Both cities could have provided better water, sanitation, and other basic resources to ensure that all citizens were protected; this would have been even easier in New York, which had a much smaller population and more time to prepare (about 185,000 people compared to 1,878,000 in London) [86,87]. As with most major cities at the time, New York faced regular water shortages, but it did not have the massive sewage problem and drinking water contamination that plagued most of the London metropolitan area. Slums and tenement houses were also much less extensive in New York.

Based on total available resources relative to the size of the problem, New York should have fared better than London in 1832. To understand why mortality per capita was much higher in New York (3.5%) than London (0.5%; see Figure 1), we need to consider problem narratives, power disconnects, and their impacts on the other two capacities: understanding and incentives. First, there was a lot of uncertainty and misinformation about how cholera was transmitted and how it could be treated. Understanding was, therefore, generally limited but people still constructed problem narratives to make sense of this threat and decide how to deal with it. The first, and most productive, narrative linked the spread of cholera to a lack of sanitation, including access to clean water. This narrative was championed primarily by doctors and was dominant in England as cholera approached. Although there were still skeptics who espoused other problem narratives, many in London believed the sanitation narrative and therefore had incentives to increase access to clean water and sewage, effectively sharing resources to increase prevention [77].

Although some advocated the sanitation narrative in New York as well, a second, counterproductive problem narrative proved more persuasive to most people with power. This "poor sinners" narrative was championed primarily by religious and political leaders who claimed that cholera was a punishment for sin. This religious version of the poor sinner's narrative allowed people to feel safe from cholera as long as they lived "godly" lives. This skewed their perception of risk from the disease, reducing the incentive to engage in prevention and widening power disconnects, while also undermining their understanding of causes and solutions. Similarly, some doctors believed that cholera could only attack people who were already in poor health, usually due to smoking, drinking, and the various other "sins" condemned by well-to-do Christians of the time. This medical version of the "poor sinners" narrative reinforced risk assessments suggesting that the poor were most vulnerable to cholera. Based in the prevailing religious beliefs of the period, this narrative dominated the discourse around cholera in New York, producing counterproductive individual responses that focused on changing the lifestyle choices of the poor rather than providing them with better access to clean water and waste removal as was the primary private response in London [88,89].

In addition, about half of New York's population fled to the countryside once news finally emerged that the disease was in the city. This highlights a class-based disconnect in which wealthy and middle-class people had a way to insulate themselves from the disease without addressing its root causes; a strategy they used before when facing other epidemic diseases like yellow fever. When London was smaller, elites used a similar tactic to try to avoid bubonic plague, smallpox, and similar diseases, but by 1832, abandoning the sprawling city was too difficult for most of its inhabitants. Unable to reduce their vulnerability through flight, Londoners had greater incentives to invest resources in improving sanitation throughout the city [78,85].

Although the city's mortality rate was much lower than other metropolitan areas affected by the pandemic, the aggregate effects of individual actions to improve sanitation in London were still not sufficient to fully prevent an epidemic. This is partly because inaccurate problem narratives still prevailed in some local municipalities, reducing incentives to share resources for prevention, but it was also because the citizens and charitable groups who subscribed to the sanitation narrative simply did not have the resources—not just financial but also technical, social, and organizational—to sufficiently clean up the city.

Private responses to complex public health problems are rarely sufficient and government resources are usually needed, especially for large polities like London or New York.

In theory, governments are expected to provide public goods when private resources are insufficient, but they may also fail to prevent public health crises because they lack sufficient financial, organizational, or legal resources. For cholera specifically, London's vast, tangled patchwork of municipal interests prevented much of the government-led collective response that could have reduced the incidence of cholera in 1832. Corruption was also a major impediment to government attempts at prevention in both cities. Politicians easily coopted government funds for their own gain and often watered-down legislation to favor landlords or others who did not want to pay the costs of ensuring decent living and working conditions for the poor [80]. For instance, New York had tried to secure sources of clean drinking water by creating a city-funded water corporation after yellow fever epidemics of the late 1700s. However, the corporation was proposed and run by a group of politicians who used it to create a bank—which was very lucrative at the time—and infamously failed in the primary task of providing clean water for decades prior to the cholera epidemic of 1832. Furthermore, local authorities regularly accepted bribes to disregard building codes and other regulations that would have reduced transmission of the disease, while politicians cared more about maintaining power and protecting their business interests than preventing the epidemic [82].

Such corruption flourished through institutionalized power disconnects resulting from a lack of accountability and transparency in both city governments. Although both cities had democratic institutions, most of the population—particularly those who were most vulnerable to cholera—were not permitted to vote and so had little political influence. This power disconnect was due to an elitist narrative which argued that only (white) "men of property," had sufficient understanding and appropriate incentives to govern. Thus, when cholera hit in 1832, only affluent white men had the vote in London, leaving the vast majority of people unrepresented in local, city, and national government. Interestingly, the Great Reform Act was passed in June of 1832, after the cholera epidemic had disspated in London. While the act did extend the franchise to middle-class men, it still excluded the majority of the population. There is also no evidence that cholera fostered the Act in any way, though both cholera and the increasingly dirty conditions in major cities may have added to middle-class demands for suffrage.

In New York, white working-class men had the right to vote, but men of color, women, and immigrants were all denied a voice. This created an important power disconnect such that, in both cities, the people who ran the government were also those who were most able to insulate themselves from the threat of cholera itself and instead felt most threatened by preventative measures such as quarantines, public provision of sanitation facilities, and the enactment or enforcement of laws requiring improved sanitation facilities in slums and tenements. Ironically, there were leaders in both cities who pushed for better sanitation and clean water but they did not have sufficient political influence to overcome the maladaptive problem narratives espoused by leaders with a vested interest in maintaining the status quo [78,88].

In short, although the private response was much more effective in London than New York, both cities failed to prevent outbreaks because government capacities were not sufficiently aligned. In London, people who controlled private resources (charitable organizations and the general public) believed a problem narrative that gave them incentives to work to prevent the disease and which pointed in the right direction to stem the tide of cholera. Nonetheless, government forces did not align because of political power disconnects and lack of organizational capacity. In New York, neither private nor public capacities aligned to support prevention, largely because people with resources (including political influence) believed in problem narratives that gave them incentives to personally avoid–rather than working to prevent–the spread of the disease in the general population.

## 4. Transition to Healthy Governance–Learning Lessons for Subsequent Crises

The previous section detailed how capacities align (or not) to switch the governance treadmill from ineffective to effective cycles of collective response. Here we consider how repeated experience with multiple crises through the cycles in the governance treadmill can generate more lasting changes that maintain the alignment of capacities to effectively prevent or mitigate future crises. For instance, after the epidemics of 1832, life for most survivors in London and New York quickly returned to "normal". There was a short period of increased charity, and a few new laws were passed, but neither city managed to fully address their sanitation problems (which got much worse), or otherwise increase their capacity to prevent future outbreaks. In addition to this crisis rebound effect, there were also maladaptive responses in each city that actually reduced capacities to prevent or mitigate additional harm during subsequent cholera epidemics in the 1840s and 1850s. It took repeated experience with increasing costs over at least three decades to build up institutions that would maintain the alignment of resources, understandings, and incentives needed to finally conquer the disease and prevent further outbreaks introduced from abroad into New York and London.

While components of healthy governance are well documented for public health issues, the process of transition to healthy governance is poorly understood across disciplines. The rest of this section explores how treadmill dynamics shape paths toward or away from healthy governance for a given public health threat. It starts by delving into the literature on healthy governance and refining our understanding of adaptive vs. maladaptive responses to a single crisis like an epidemic (Section 4.1). We then explore the barriers to healthy governance that arise from broad, pre-existing problem narratives and power disconnects that favor maladaptive responses either during a crisis or in its aftermath (Section 4.2). Third, we show how compounding issue-specific crises can create windows of opportunity for the large-scale changes in the precursors for healthy governance (Section 4.3) in the face of public health threats. Finally, we consider how scope and scale affect our analysis, with particular emphasis on how responses that are adaptive at the local scale may be maladaptive globally.

### 4.1. Components of Healthy Governance

Achieving healthy governance requires substantial changes in social, economic, and political institutions at multiple levels and scales. This point is supported by both academic and applied work in the field of public health. For instance, the Health in All Policies (HiAP) initiative emphasizes that for long-term effectiveness, health, and wellbeing should be key components of policy development across sectors and scales of government, including policies associated with education, employment opportunities, and social justice [3,90,91]. The WHO and other international bodies have been pushing for HiAP for at least 10 years, but progress has been slow and the concept remains largely aspirational [92]. This effort is indicative of many similar attempts to move toward healthy governance in other policy fora, such as the UN Sustainable Development Goals [93].

Similarly, scholarly research from multiple disciplines suggests that healthy governance for any type of issue depends heavily on the broader social, economic, and political processes that collectively determine access to and allocation of resources. In particular, resources need to be distributed equitably to guarantee that all groups will be able to respond effectively to every potential crisis. This may require a change in broader social, political, and economic institutions to narrow power disconnects by shifting resources to people who have incentives to respond productively. Similar conclusions are drawn from research on the Social Determinants of Health (SDoH) [32,94], sustainability science [95–97], development economics [98–100], environmental justice [35,72], collaborative governance [101–103], human rights action [14,104], and political science more generally [30,105,106]. With respect to the SDoH specifically, institutional changes should alter access to and representation within agriculture and food production/distribution, education, employment, water and sanitation, healthcare services, and housing, among many others. Groups with greater

access to these resources may have more power to control those institutions that affect resource distribution. However, processes of building social capital are complex, reflecting nested hierarchies of individuals and groups within larger geopolitical and spatial contexts [107]. In the case of cholera, for example, the macro-level context (global neoliberalism and populism) may be viewed as inhibiting social capital and hindering disease prevention in the low-income countries, while simultaneously perpetuating social inequities associated with health disparities in higher income countries.

Combining concepts from Sections 2 and 3 with these insights from the literature, we can refine our definition of adaptive collective response to encompass the creation of institutions that maintain an alignment of capacities toward effective response in spite of the crisis rebound effect. Adaptive responses generally include improving signals (e.g., better monitoring of disease pathways), shifting problem narratives to focus on causes rather than effects (e.g., targeting the disease not the symptoms), and empowering people who have incentives to respond quickly and effectively (e.g., increasing access to sanitation, increasing government accountability to the poor, increasing representation of marginalized populations). Maladaptive responses, in contrast, are those that misalign capacities, allowing the governance treadmill to get stuck in ineffective cycles, with few and temporary improvements in effectiveness. Responses that dampen problem signals, promote inaccurate problem narratives, or widen power disconnects tend to be maladaptive. Both types of responses can occur before, during, and after a crisis and adaptiveness may interact with effectiveness in surprising ways, as short-term coping responses that are effective when preventing one crisis may reduce the potential for effective responses to future crises.

We can illustrate these points using the cholera examples. On one hand, the poor sinners narrative was maladaptive as well as ineffective because it reinforced the existing power disconnects between the rich and poor by suggesting that the poor were undeserving of resources that were concentrated with the rich. After the outbreaks of the 1830s, an altered version of this narrative was leveraged by anti-immigrant political parties that blamed new arrivals for the disease and used the power that they gained thereby to actively repress social movements aimed at empowering the working poor. Irish immigrants were particularly targeted, in part because they were the first to be diagnosed with cholera in New York, but also because of deeply entrenched ethnic and religious prejudices as well as fears that the Irish were gaining political and economic influence. Once the initial anger over the outbreak subsided, anti-immigrant influence declined as well, and more moderate parties returned to power [85]. Ultimately, this served the purpose of elites at the time, who were happy to scapegoat immigrants, drawing attention away from their own culpability for the outbreak and draining political will for real reforms that would narrow power disconnects over the long term.

Although it engendered more effective collective response in 1832, the sanitation narrative in London proved to be maladaptive by omission. While it pointed charities in the right direction to effectively reduce the spread of the disease, it did not address the fundamental inequities, market failures, and government ineffectiveness that created the conditions for epidemic cholera. Indeed, the "sanitation movement" that swept through London after 1832 was largely ineffective at preventing additional outbreaks precisely because it did not address these issues. Some activists pushed for better health and sanitation regulation, but broader reform was not on the agenda. Rather than narrow power disconnects by shifting resources like money and influence toward those most vulnerable to water-borne diseases (for instance by supporting labor movements, extending voting rights, or demanding better social services), wealthy and middle-class Londoners instead invested in private water and sewage services, widening power disconnects by insulating themselves from the perceived risk of future outbreaks. Well-off New Yorkers took a similar approach, though they were better able to leverage government funding to pay for clean water and improved sanitation in their neighborhoods.

### 4.2. Barriers to Healthy Governance

Early responses in both London and New York proved to be maladaptive because of strong barriers to changes in problem narratives and power disconnects. Some of these barriers were overcome by repeated experience with multiple cholera epidemics in each city. Others were only overcome with larger changes in the social, economic, and political systems that shaped responses to cholera. In reality, these shifts are not separable, as there were also feedbacks between experiences with cholera and broader social transformations. In this section, we examine the three types of interconnected barriers to healthy governance summarized in Figure 5: (1) psychological (heuristics and biases), (2) informal institutions (norms), and (3) formal institutions (laws). We will examine the effects of cumulative experiences with similar crises and the importance of broader social change for overcoming these barriers in the next sections.

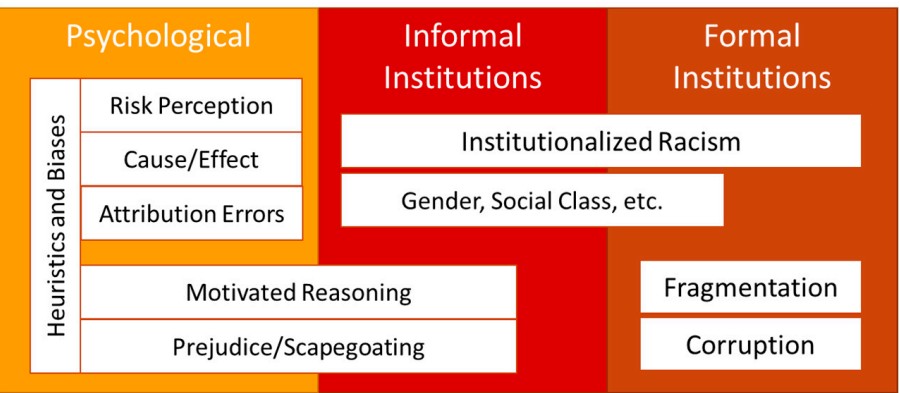

**Figure 5.** Barriers to Healthy Governance.

First, there are a number of psychological barriers to healthy governance. People do not change their problem narratives easily. This is doubly true when they have incentives to maintain particular narratives. For example, even before cholera hit New York there was ample evidence against the poor sinners problem narrative, but believers exhibited confirmation bias, or the tendency to accept information that supports prior beliefs and reject information that contradicts prior beliefs, regardless of its veracity [76,108]. There are many other biases and related heuristics, or mental shortcuts, that can prevent learning from taking place, before, during, and after a crisis [92,109,110]. Although we cannot go into detail here, these psychological factors affect our ability to perceive risk [111], to accurately assess causal relationships [112], and to honestly assess the effects of our own decisions or those of others [113], thereby undermining the development of adaptive problem narratives. Similarly, people frequently use motivated reasoning to justify narratives that bestow important psychological benefits, such as validation of one's position in society and rationalizing benefits from the inequitable distribution of resources [56,114,115].

Psychological barriers are also shaped and reinforced by informal socioeconomic and political institutions that keep power disconnects wide. These have been studied in several literatures and have been shown to undermine the long run effectiveness of public health efforts. For example, Jones [29] provides a framework for identifying the structural causes of "race"-associated differences in both communicable and non-communicable diseases within a larger socio-political context. She defines institutionalized racism as "differential access to the goods, services, and opportunities of society by race; a form of racism that is "normative, sometimes legalized, and often manifests as inherited disadvantage . . . having been codified in our institutions of custom, practice, and law, so there need not be an identifiable perpetrator. Indeed, institutionalized racism is often evident as inaction in the face of need" [1,29,116,117].

We see the effects of racism, classism, and religious prejudice clearly in the examples of cholera response in London and New York as well. While these factors were more obvious in the persistence of the poor sinners and anti-immigrant problem narratives in New York,

they also helped shape the paternalistic components of the sanitation narrative and were used to rationalize insulating and counterproductive approaches to sanitation after 1832. Indeed, as the sanitation problem in the city grew, well-off Londoners used social class to rationalize illegally breaking into London's floodwater control system to dump their sewage, and otherwise cleaning up their own communities by transferring their waste to less affluent areas.

As noted by Jones [29] and others, laws and other formal institutions may reinforce power disconnects and therefore may have to be changed in order to transition to healthy governance [1,118–121], though such change is particularly difficult when powerful actors derive influence from the arrangement [122,123]. Most clearly in the examples, laws restricting voting to a relatively small group of privileged elites contributed to the prolonged ineffectiveness of response to cholera in London and New York. Legal and bureaucratic structures also created vested interests that resisted attempts at reform, as people who wielded even small amounts of power (e.g., local water board, sanitation inspectors) organized to resist changes that would reduce their influence or decrease opportunities for bribes or other political rents. The fragmented structure of government, with different resources controlled by different groups at different levels of analysis also prevented effective response in both cities, at least until the state (New York) or national (UK) governments were finally persuaded to take action that would harmonize response across municipalities. This returns us to the fundamental problem of building up enough political will to shift collective response toward healthy governance. We turn to this process in the next section.

*4.3. Precursours of Healthy Governance*

Despite the barriers described above, transition to healthy governance is possible, at least for some public health threats. According to literatures in sociology and political science, by increasing attention to an issue, human security crises like epidemics can create conditions for just such a transformation. However, this literature also suggests that the windows of opportunity associated with a specific crisis tend to be narrow and they often only lead to short-run improvements in effectiveness as described above [124–127]. Other authors also show that incremental change can be more lasting than the episodic change associated with crisis response, although this finding is contested [128,129]. Contemporary social entrepreneurship frameworks such as Collective Impact 3.0 suggest that conditions such as identifying a collaborative vision of a 'problem', shared measurement systems, mutually reinforcing strategies, continuous and open communication, and establishing a 'backbone' support organization is important [130]. The World Health Organization [8] suggests that healthy governance will require a greater sense of shared purpose among institutions and sectors, as well as a greater global perspective in public health education and training.

We argue that the transition to healthy governance depends on both incremental change and learning from repeated experiences with crisis, which may change the state of the system in relation to subsequent disruptions. The main question, then, is whether or not both gradual and episodic pressures are reinforcing the barriers described in the previous section or building up the precursors for healthy governance described in Figure 6. In particular, the question is whether or not gradual AND episodic changes are realigning capacities toward healthy governance by narrowing power disconnects and popularizing adaptive problem narratives. While Figure 6 represents each precursor separately, there are many feedbacks between them, such as when declines in prejudice-based problem narratives empower vulnerable groups or increasing reliance on evidence-based narratives foster the development of easier and more effective response options. Rather than enumerating each of the components of Figure 6 separately—and trying to track all of the theoretical interactions between them—the rest of this section traces the recursive process of transition through changes in these precursors for our representative examples of cholera response in London and New York. This analysis may form the basis for a more synthetic approach in future work.

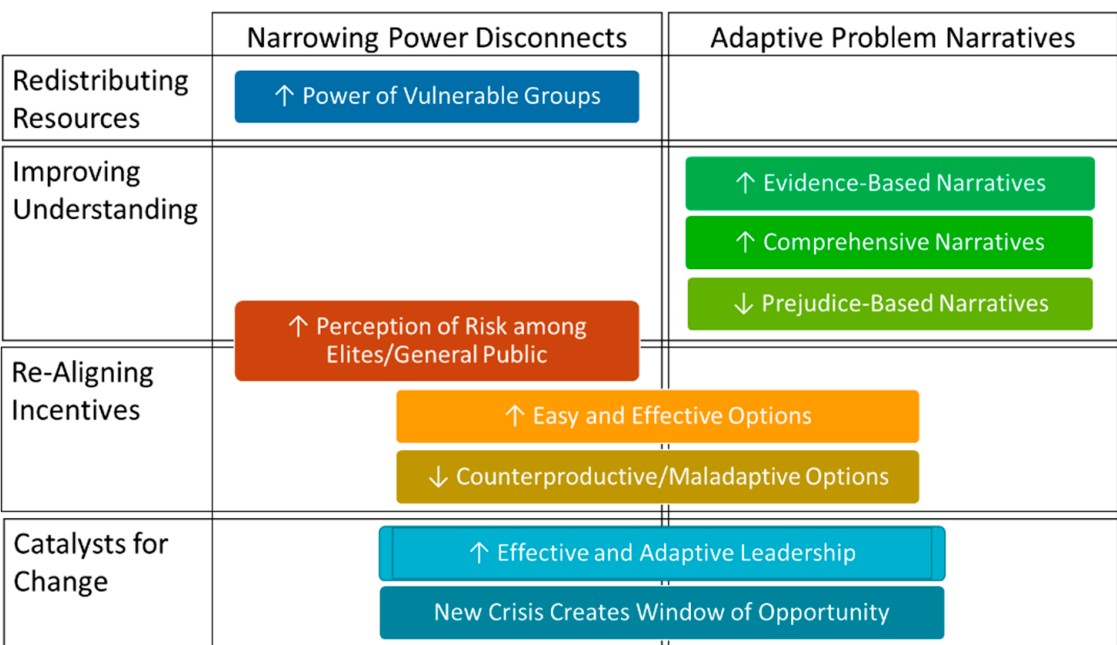

**Figure 6.** Precursors to Healthy Governance.

First, let us consider changes prompted by repeated experience with the same type of public health threat (e.g., multiple ineffective cycles in the cholera treadmill). In general, repeated experience can align understanding and incentives by increasing public concern about a given human security threat (↑ Perception of Risk in Figure 6), though this depends heavily on the type of threat. People often develop psychological coping mechanisms when grappling with chronic or endemic health threats, which may result in reduced concern through fatigue with repeated exposure (see psychological barriers above). However, for the periodic, epidemic threats described here, we can expect that repeated experience will generally increase threat perception [76]. For instance, from the historical literature we know that people in both New York and London believed that the cholera epidemics of the 1840s and 1850s were worse than the epidemic of 1832, because death tolls were higher (even though per capita death rates were lower due to population growth and metropolitan expansion away from polluted water sources). In addition, with only five years between these severe epidemics (1848–1849 and 1854–1855) and the longer duration of risk as the third pandemic lasted 14 years, people perceived that cholera was a persistent problem, rather than a short-term threat.

Repeated experience with crisis can also increase understanding through observation, trial and error, and scientific advances (↑ Evidence Based Narratives). This includes learning about the ineffectiveness of counter-productive individual responses or maladaptive collective responses (↓ Counterproductive/Maladaptive Options; ↓ Prejudice-Based Narratives). Interestingly, the latter can be more important in some cases. For instance, even though John Snow used the famous "water pump experiment" to verify the theories about water-based spread of cholera during the 1854 epidemic, his work was not accepted for several decades. Most people continued to believe that bad smells or clouds (miasmas) transmitted diseases, though the sanitation narrative persisted because many agreed that sewage and other forms of "filth" could be a source of disease-bearing miasmas. In contrast, when news broke that private water companies were providing water drawn just downstream from sewage outfalls to wealthy London residents, they took this as a clear lesson that private, market-based responses were not working. Although this "crisis" was not directly linked to mortality associated with a cholera epidemic it occurred during the ever-looming threat of the third pandemic (1846–1860). Furthermore, in discrediting maladaptive responses to previous outbreaks this "crisis" reinforced perceived needs for

government provision of sanitation, underscoring the importance of the 'assurance' core function as it is known in contemporary public health practice (↑ Comprehensive Narratives). This also increased elite pressures on policy makers during the 1850s in order to protect the population from this threat (↑ Perception of Risk).

This brings us back to changes that occur between crises like epidemics. First, while a majority of the public might be subject to the crisis rebound effect, smaller groups may continue to work to align capacities for public health between crises. In our examples, these groups developed detailed problem narratives that linked the persistence of cholera and other diseases to broader governance failures and pursued long-term strategies to fix the broken system (↑ Comprehensive Narratives; ↑ Easy and Effective Options). In both cities, doctors who believed in scientific approaches to medicine (a relatively new idea at the time) created professional organizations and lobbied governments to adopt proactive approaches to public health. Engineers developed new technologies that made provision of clean water and sanitation services economically and politically feasible, even for large cities like London and New York. Ambitious politicians and bureaucrats built up new power bases by taking advantage of the increasing demand for reform, building trust with vested interests, and developing policies that finally aligned incentives with resources and understanding over the long run (↑ Effective and Adaptive Leadership).

Second, larger-scale changes in social institutions and world views that reinforce most of the barriers described in the previous section can be fostered by experience with multiple, complementary public health threats (↑ Perception of Risk; ↑ Comprehensive Narratives). In both London and New York, cholera was not the only public health threat nor, indeed, was public health the only source of instability. Both cities experienced multiple crises as well as escalating low-grade costs from the maladaptive narratives that kept power disconnects wide and undermined effective governance. By the 1850s, typhus, tuberculosis, and many other diseases were endemic in both cities, and growing populations of working-class poor also faced chronic malnutrition, high infant and child mortality, and numerous other health threats. Smoke and pollution from factories built up along with the sewage and garbage from household use, spreading pollutants over both cities. Financial crises periodically swept through as speculative bubbles burst. Crime was high. Fires were a constant threat.

At the same time, elites found that they could no longer insulate themselves from the many problems of the age (↓ Counterproductive/Maladaptive Options; ↑ Perception of Risk). Risk from disease, crime, and the other problems described above overwhelmed the palliative responses of the past. Between bouts with cholera, elites also found that they could not avoid the growing filth and stench of the city—or its related diseases. The scandals about private water companies using polluted sources in London were already described above. In New York, corruption was revealed by growing water shortages and, as the city grew rapidly, it became much more difficult to flee to the countryside to avoid disease. New York State officials reinforced the latter lesson by banning travel out of the city during cholera outbreaks in the 1850s, having learned that exodus from the city spread the disease to the rest of the state during the epidemics of the 1840s.

Third, socioeconomic and political changes fostered a more equitable distribution of resources, particularly shifting political and economic power to the people who were most at risk from disease (↑ Power of Vulnerable Groups). This narrowing of power disconnects was not a response to cholera alone but also to the many negative effects of poverty and destructive governance. Poor and politically marginalized people who were most vulnerable to cholera and other diseases found ways to gain political and economic influence. Most of the resulting systemic changes were gradual, such as the rise of the middle class, the empowerment of workers through labor unions, and (in New York) political organization among second generation immigrants. As these groups gained influence through size and organized collective action, they demanded and received changes to formal institutions. These changes further narrowed power disconnects by increasing democratic representation and accountability in government (↓ Counterproductive/Maladaptive Options).

Of course, many elites doubled down on maladaptive narratives that blamed the poor for all of the problems in each city, but their voices were less persuasive as new groups gained power. For instance, the poor sinners narrative fell out of favor in part because the newly powerful lower and middle classes had experienced poverty (↓ Prejudice-Based Narratives). They knew its limits and understood its causes much better than even the best-intentioned members of the upper classes. For many, this translated into a commitment to improving conditions and opportunities for the poor. The growth of positivism as a dominant world view also helped to oust the "poor sinners" narrative and other metaphysical explanations of disease (↑ Evidence-Based Narratives). This was largely a generational change, as younger people were more willing to accept the (then) radical idea that physical evidence mattered more than metaphysics [131]. Indeed, most of the innovations that made solving the problem of cholera feasible in the mid-1800s came from leaders and entrepreneurs who were children or young adults during the 1832 epidemics. Interestingly, most of these individuals came from poor or middle-class backgrounds as well (↑ Power of Vulnerable Groups) [79–81,84,85].

In both cities, transition to healthy governance did not occur until windows of opportunity were opened during subsequent crises. In London, a heat wave sent "the Great Stink" into the city from the Thames in 1858. The overpowering stench of sewage was blown directly into the Houses of Parliament. Most politicians—like the public they served—believed that this terrible miasma could transmit diseases directly and, while cholera was not a threat at the time, they still feared typhus and other diseases of the period. Under the leadership of politicians like Sir Benjamin Hall (a Welsh civil engineer before he became a politician) and bureaucrats like Sir Joseph Bazalgette (an engineer and entrepreneur) the crisis of the Great Stink gave the sanitation movement the momentum it needed to push through construction of a massive city-wide sewage system to clean up the streets, the Thames, and other sources of water around the city (↑ Effective and Adaptive Leadership; ↑ Easy and Effective Options). While this feat of engineering is usually given all the credit for ending cholera and other waterborne diseases in London, the Great Stink also opened a policy window for other long-sought reforms. Hall, who was President of the Board of Health during the previous cholera outbreak, used it as an opportunity to push for health care reform, including national medical licensing and the replacement of local medical boards with a national system. He also worked with local leaders to overhaul the municipal management of water and sewage, shifting bureaucratic incentives toward maintaining provision of clean water and sewage and away from taking bribes from absentee landlords [78–80].

The solution to the cholera threat was different in New York, and it came almost a decade later. Cholera was endemic in the city's slums for most of the 1850s and early 1860s. However, as the third pandemic approached New York in 1865, there was a massive public outcry to improve sanitation and otherwise protect the city from another epidemic outbreak of the disease (↑ Power of Vulnerable Groups; ↑ Perception of Risk). Unfortunately, this was insufficient to overcome formal institutional barriers in New York City. New York state authorities finally over-ruled city government, legislating a strong city-wide preventative health system right on the cusp of the 1866 outbreak (↑ Easy and Effective Options). This move was initiated partly because of fears that the epidemic would yet again spread to other cities (↑ Comprehensive Narratives), partly because many, mostly younger doctors who espoused the sanitation narrative had organized in order to seek influence over public policy at the state level (↑ Effective and Adaptive Leadership), and partly because state authorities had a broader agenda to increase their control over the city (↓ Counterproductive/Maladaptive Options). This power shift from city to state set the stage for the "battle" against cholera in 1865, which was led by those same "pro-science" doctors, who were appointed by the state and given sweeping powers to appropriate resources to clean up slums and enforce existing sanitation laws, as well as to monitor people coming into the city, treat them effectively, and, if necessary, quarantine them

(↑ Power of Vulnerable Groups) [82,84,85,132]. This narrative shift was crucial in responding locally to global human security threats.

From the above examples, we can see that there is not a linear progression in the transition to healthy governance. Each cycle of the treadmill brings opportunities to overcome barriers and reset the precursors to governance. This includes changes that occur between crises, in latent phases of the treadmill, when people have a chance to foster more gradual and sweeping systemic changes such as inventing easier yet effective solutions to a problem (e.g., Bazalgette's sewage innovations, Hall's political entrepreneurship) and organizing to demand greater political or economic influence. Problem narratives, too, may change between crises, through learning and generational shifts. Most important, reinforcing feedbacks between more adaptive problem narratives and narrower power disconnects can lead to adaptive transitions, particularly when catalyzed by effective and adaptive leadership at the beginning of a new cycle of the treadmill.

### 4.4. Scope and Scale

The last area to examine in transitions to healthy governance is the importance of scope and scale. Intuitively, public health threats that are spread over a larger area and/or that affect more people usually require more resources for prevention, all else equal. On the other hand, when risks are widespread it may be easier to mobilize resources because risks are more evenly distributed, which narrows power disconnects by giving the people who control resources an incentive to use them more equitably. Our point here is more nuanced, however. Assessing the transition to healthy governance depends heavily on the scope of the threat and the scale of analysis. In other words, responses that are adaptive for one type of health threat in one geographic location may be maladaptive for other types of health threats and/or other areas of the world. Figure 7 highlights this concern and shows how local-level treadmill dynamics can either lead to adaptive or maladaptive global dynamics depending on the impacts on power disconnects, problem narratives, and response options (paralleling Figure 6).

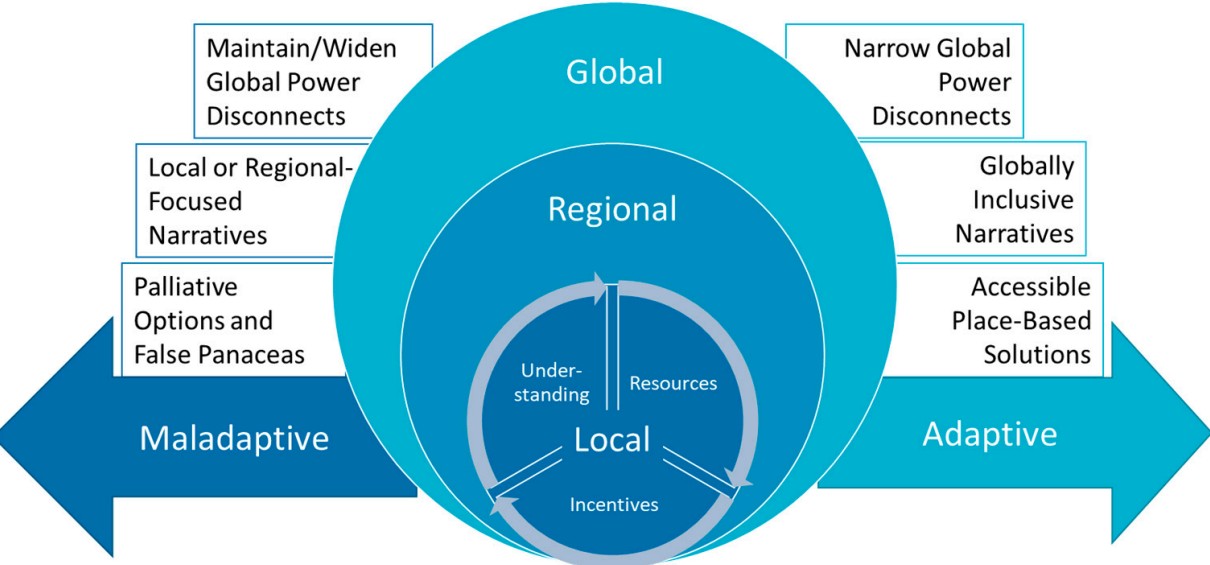

**Figure 7.** Precursors to Governance Across Scales.

For instance, winning the battle with cholera in New York, London, and other resource-rich areas also opened the door for the crisis rebound effect at other levels and for other issues. By syphoning off public attention and political will, this 'victory' failed to maintain alignment for prevention in resource-poor areas locally and globally. Locally, preventing cholera widened power disconnects because the well-to-do again felt sufficiently insulated from "diseases of the poor" including tuberculosis, malnutrition, dysentery, and typhoid.

On the local scale, unequal territorial distribution of exposure to risks and illness in certain social groups, is called environmental injustice [133], or many times, environmental racism [72]. These populations are not only more affected, but they also have less responsive capacity. For instance, conditions for the poor in London, New York, and other wealthy polities remained unhealthy well into the 20th century even if cholera was no longer a threat [78–82,84,85].

Globally, while cholera, typhoid, and similar water-borne diseases were uncommon in most high-income countries by the end of the 19th century, they remain endemic in many low-income countries, causing an estimated 21,000–143,000 deaths annually [134]. Furthermore, although the last cholera pandemic ended in 1977, the disease can still reach epidemic proportions as shown by the outbreaks in Haiti from 2010–2019 [135] and Yemen [136]. Thus, effective prevention of cholera in high-income countries turned out to be a maladaptive response from the global perspective, as it greatly reduced elite perceptions of risks (widening global power disconnects) and thereby also reduced incentives to share resources internationally to entirely eradicate the disease.

Perceiving the permanence of these inequalities, authors such as Dryzek and Pickering [137] provoke debate on the need to think about environmental justice on a planetary scale; a planetary justice that goes beyond national borders, across generation and also for non-humans. In the same direction, Kashwan et al. [138] highlight the need to "prioritize the poor in earth system governance", recognizing an imbalance of forces, reactions, and demands on a planetary scale: "Rapid planetary-scale processes have reinforced and further created vast injustices at international, national, and subnational levels." During COVID-19, this was noticed a consistently, especially concerning vaccination—from production, marketing, and distribution. Poorer nations, mainly in Africa and Latin America, took longer to start vaccinating and have more difficulties in acquiring it [139,140]. For the cholera cases in London and New York, as well as COVID-19, as long as power disconnects remain wide and problem narratives continue to prioritize national/regional health above global health, healthy governance at the global scale is not likely.

The bigger picture is still encouraging, however. Although the global health is still mired in a complex, multi-level treadmill of responsive governance—and destructive governance can be observed for some issues in some areas—there can be little doubt that the nested treadmills for cholera and many other diseases have increased our preparedness to respond to new public health threats through multiple changes in the three capacities described in Section 3. For example, one concrete advance was in internalization of the sanitation narrative among medical providers, policy makers, and the public—not just in London and New York, but throughout the world [94,141,142]. In addition to this non-regime [143], experience with the cholera pandemics also fostered formal international cooperation in the form of the International Sanitary Commission (a precursor of the World Health Organization), which dealt exclusively with cholera from its inception in 1851 until the end of the century, when attention switched to other diseases that seemed to pose greater risks to high-income countries [144]. Once this new international health governance system was established, it faced similar constraints in terms of evolving to address new threats and conditions [145,146].

More general changes include improved health care through the professionalization and democratization of medical science, increased public trust in health care providers due to improved outcomes, increased public pressure on governments to provide place-based resources for public health, and technological innovations pioneered by inventors in both the public and private sectors. As described by the WHO [8], strategies that demonstrate the co-benefits of a public health intervention to multiple sectors are key, as well as identifying perceived "win-win" scenarios that may facilitate cost-sharing. For example, investments in water and sanitation systems not only reduce the risk of water-borne disease but can also create infrastructure that is more resilient to extreme weather events associated with climate change. The 2010 Adelaide Statement on Health in All Policies calls for "incentives" and "budgetary commitment" to help public agencies work

together and share resources for these types of integrated solutions [147]. In response to COVID-19, we also see increased cooperation between levels in the Global South, including city networks and transactional interaction between social actors [140]. This is evidence that countries in the Global South are organizing to empower themselves, a key step toward narrowing global power disconnects to foster transition to healthy governance world-wide.

## 5. Conclusions

Although there have been many advances in public health since the cholera cases described above, many barriers remain with respect to effective disease prevention. Recent literature has described "Public Health 3.0" as a renewed commitment to addressing the social determinants of health [148], and more recent scholarship has described "public health reimagined" [4] in the wake of COVID-19 as an evolving strategy that will need to adapt and respond to changing health and social needs. Public health professionals must continue their work to alter problem narratives through education and outreach, but they need to work alongside policymakers, advocates, entrepreneurs, engaged citizens, and other interdisciplinary scientists to frame information in ways that resonate with a more diverse set of stakeholders. This includes utilizing more targeted, accessible communication strategies [149] and integrative policy platforms [150] to narrow power disconnects through institutions that empower vulnerable populations by shifting resources and authority. Additionally, this includes facilitating globally accessible place-based solutions while engaging as global citizens to advance planetary justice.

To achieve this next step toward healthy governance, public health professionals will need support from policy makers, businesses, grass-roots organizations, and the public. Better collaboration and reframing information may help with this, but ultimately resources must be aligned with globally inclusive narratives. In ordinary times, barriers to changes in problem narratives and power disconnects are strong, which makes it hard to transition out of ineffective patterns for specific threats and even more difficult to change the larger social context that reinforces those issue-specific barriers. Gradual and small-scale changes are possible, but lack of public attention and resistance from powerful vested interests tends to prevent the systemic changes prescribed by public health [151] and environmental [152] experts.

However, much like Londoners and New Yorkers in the 1850s and 1860s, we are living in interesting times, where a confluence of crises has created substantial pressures for systemic change in cities, countries, and international organizations all around the world. COVID-19 has concentrated attention on public health policies while at the same time revealing larger fissures in political and economic systems at different levels of analysis. On the one hand, powerful people are waking up to the dangers and risks inherent in the current system. On the other hand, vulnerable and previously marginalized populations are organizing and empowering themselves through the global #BlackLivesMatter; Black, Indigenous, People of Color (BIPOC); Climate Marches; and similar international movements [133,153,154]. Thus, the window of opportunity to shift closer to healthy governance is wide open [94,150].

Whether we take this opportunity to achieve healthy governance globally depends on the detailed machinations of the treadmill across levels of analysis, but some broad conclusions can be drawn. First, as we have already seen, response will vary around the world and countries that already have capacities aligned toward effective epidemic disease prevention are generally faring better than those that lack resources, are conflicted in their understanding of methods of prevention, or are controlled by powerful people who believe that they and those they care about are not vulnerable to the disease [36,155,156]. At one extreme, COVID-19 combined with violent conflict is contributing to full-scale systemic collapse like in Yemen [157]. International power disconnects are also shaping our global response. Access to PPE, vaccines, and other resources is unequally distributed, and we are already seeing wealthy countries hoarding some key resources and even using those resources as leverage in international relations [158–160]. In developing countries that are undergoing epidemiologic transition (in which rates of communicable diseases remain

high, and chronic diseases rates are also rising) vaccines can be difficult to deliver and the chronic disease burdens often remain unaddressed. In countries like the US, ideological disputes, power disconnects, and institutionally determined incentives are undermining prevention efforts by policy makers and the public alike [150,161]. Much like London and New York in 1832, it may take multiple experiences with crises—along with gradual changes in the alignment of capacities before these countries attain healthy governance that can tackle future threats. As with cholera, failure to address these threats in low-income countries may prolong the harm—including economic damage—in wealthy nations as well [162]. Power disconnects at multiple levels are key factors for scholars to consider in understanding human security threat response within and beyond borders.

Second, while the windows of opportunity created by crises can be important turning points in governance, the treadmill analogy and cholera examples show us that what happens in the aftermath and in between crisis events matters as well. If the crisis rebound effect kicks in, then gains made toward healthy governance can easily be undermined by vested interests, such as the city inspectors, slum lords, and private water and sewage companies that found their way around legislation passed post-cholera in both London and New York. One way to counteract this process is to change formal institutions, creating or reforming government agencies to increase accountability when there are crisis-driven windows of opportunity. However, these institutions, too, tend to erode if broader social norms remain unchanged. Ideally, the public will internalize new problem narratives that provide incentives to hold governments, businesses, and others to account for the provision of needed place-based public health services. In the cholera examples, this meant both the internalization of the sanitation narrative and the long-term empowerment of vulnerable populations (middle class in London, Irish, and other immigrants in New York). As such, exploring the relationships between problem narratives and power disconnects—ideas and incentives—is still a much needed third way in all of the disciplines we draw on here [163].

Collectively, these insights reinforce the idea that healthy governance is an ongoing process that requires constant attention, rather than a series of isolated solutions or fragmented responses to crises. As focusing events, pandemics can create windows of opportunity, during which stakeholders may be more willing to change their problem narratives and when political and economic power may briefly become more fluid. As scholars, we need to expand interdisciplinary research into the effects of crisis and response on government success or failure at multiple levels of analysis. This could include further research on how problem narratives and power disconnects create different patterns of governance over time but might also extend to identifying other factors that alter the precursors to governance. As a global community, we need to work consistently and intentionally to maintain hard-won advances in system transformations that can support human wellbeing.

**Author Contributions:** Conceptualization, D.G.W.; methodology and data curation, D.G.W.; investigation, D.G.W., M.A., S.A.A., R.S.W. and J.A.H.; writing—original draft preparation, review and editing, D.G.W., M.A., S.A.A., R.S.W., J.A.H., L.S., A.L.P., P.H.C.T., A.A. and O.Y.; visualization, D.G.W. All authors have read and agreed to the published version of the manuscript.

**Funding:** This work was supported by the National Socio-Environmental Synthesis Center (SESYNC) under funding received from the National Science Foundation, grant number DBI-1052875. The São Paulo Research Foundation (FAPESP) also funded this research, grant number 2018/06685-9.

**Institutional Review Board Statement:** Not applicable.

**Informed Consent Statement:** Not applicable.

**Data Availability Statement:** All data are available from sources cited herein.

**Conflicts of Interest:** The authors declare no conflict of interest.

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
