# Peer review of "Learning from the Past: Pandemics and the Governance Treadmill"

_sustainability, doi:10.3390/su14063683_

Round 1

Reviewer 1 Report

Congratulations the idea of problem narratives in relation with the capacity of evolve understanding, resources and incentives is very well accomplished. 

The main objective is very original as well as interesting. The paper is very well written and with a deep understanding and analysis of the past to foster the conditions of posibility for HiAP or Public Health 3.0. 

The theadmill is very well explained theoretically and empirically, and the idea of narratives and power disconnects is very well analised. 

The only question I think it could be better explain is how to scale up from a very local and historical examples to the global context we need to take into the account nowadays, as we have experimented with Covid19. The global dimension of the Health governance should have been helped to draw a new model of treadmill, in order to include individual and institutional responses influenced by all scales narratives as much as global, national and local narratives, resources and capacities. 

Author Response

Many thanks to this reviewer for their helpful comments. In response to their suggestion,

“I think it could be better explain is how to scale up from a very local and historical examples to the global context we need to take into the account nowadays, as we have experimented with Covid19. The global dimension of the Health governance should have been helped to draw a new model of treadmill, in order to include individual and institutional responses influenced by all scales narratives as much as global, national and local narratives, resources and capacities.”,

we added some new text and a figure to Section 4.4 on Scope and Scale and to the Conclusion. Figure 7 summarizes how changes in local-level treadmills can potentially influence regional and global treadmills in adaptive or maladaptive directions. Additional text describes some of the difficulties that might need to be overcome to get to healthy governance at the global level (drawn largely from the literature on environmental justice) as well as some of the strides that are currently being made in that direction (e.g. cooperation within the Global South). It is a complement to the newly added Figure 6, which helps to relate the precursors to governance back to changes in capacities, problem narratives, and power disconnects through treadmill dynamics. That said, we recognize that a full exploration of this issue is not feasible within the scope of this paper and so also dialed back some of our claims, as evinced by the new title, Learning from the Past: Pandemics and the Governance Treadmill.

Reviewer 2 Report

The paper presents an approach based on the concept of the governance treadmill to demonstrate cross-level dynamics that help or hinder the alignment of capacities toward prevention during public health crises, such as COVID-19.

Most of the examples and arguments discussed in the paper are based on the analysis of the cholera epidemics in London and New York in the 1800s.

However, the paper title seems to suggest it reflect on Transnational Health Governance as if it were to focus on the international relations of different countries when tackling those issues.  Instead, most of the discussion is centred on comparing differences in the local approaches used in the two cities.  I believe the paper has important discussions as it is but it should make it clearer to readers what they can expect.

Despite having many graphical representations of models and concepts up to Section 3, Section 4 is very dense on text and has very little synthesis of the concepts presented in the same fashion as the previous sections.

The paper is generally very well written.  

On line 51, I believe it would be beneficial to transcribe the development goals mentioned in the text to make reading easier.

Line 73: that that (typo)

Author Response

We greatly appreciate Reviewer 2’s comments and enumerate our responses to each of their points below.

  1. the paper title seems to suggest it reflect on Transnational Health Governance as if it were to focus on the international relations of different countries when tackling those issues.  Instead, most of the discussion is centred on comparing differences in the local approaches used in the two cities.  I believe the paper has important discussions as it is but it should make it clearer to readers what they can expect.

We changed the title to Learning from the Past: Pandemics and the Governance Treadmill. We also dialed back some of our claims in the introduction.

  1. Despite having many graphical representations of models and concepts up to Section 3, Section 4 is very dense on text and has very little synthesis of the concepts presented in the same fashion as the previous sections.

Section 4 was meant to be more analytical and less synthetic, but to address the reviewer’s very reasonable concerns we added a figure with requisite explanatory text to Sections 4.2, 4.3, and 4.4.

In Section 4.2, we summarized the barriers to transition to healthy governance using a structure similar to Figure 4 on capacities for effective response.

Section 4.3 required a bit more work. We changed the title of the section to “Precursors of Healthy Governance” to better reflect the content of the section. We then created a matrix that relates the precursors covered in the section back to the concepts of capacities, problem narratives, and power disconnects presented in Section 2. Windows of opportunity (the original section title) were folded into the concept of precursors to governance in the matrix.  Unfortunately, there is not room in this paper for a full synthetic approach to this process of transition, but we referenced this matrix throughout the rest of the section to better show how repeated experience with extreme events can lead to changes in problem narratives and power disconnects that set the stage for healthy governance. We also went back through the rest of the text to slightly realign the discussion to lead toward this section on precursors. Lastly, we note in the introduction to the section that the main point is analytical and that much future work will be required to develop a more complete understanding of these dynamics. 

Our changes to Section 4.4 were mainly in response to Reviewer 1’s comments but the addition of Figure 7, which illustrates the cross-level implications of responsive governance should provide readers with an additional guidepost to this section of the text.

  1. On line 51, I believe it would be beneficial to transcribe the development goals mentioned in the text to make reading easier.

Done

  1. Line 73: that that (typo)

Done

We also copy-edited several times, catching other small typos in the text and in the figures.